∂ | Open Peer Review | Antimicrobial Chemotherapy | Research Article

# *In vitro* antibacterial activity of dinuclear thiolato-bridged ruthenium(II)-arene compounds

Quentin Bugnon,[1,2] Camilo Melendez,[2] Oksana Desiatkina,[2] Louis Fayolles de Chaptes,[2] Isabelle Holzer,[2] Emilia Păunescu,[2] Markus Hilty,[1] Julien Furrer[2]

**ABSTRACT** The antibacterial activity of 22 thiolato-bridged dinuclear ruthenium(II)-arene compounds was assessed *in vitro* against *Escherichia coli*, *Streptococcus pneumoniae*, and *Staphylococcus aureus*. None of the compounds efficiently inhibited the growth of the three *E. coli* strains tested, and only compound **5** exhibited a medium activity against this bacterium [MIC (minimum inhibitory concentration) of 25 µM]. However, a significant antibacterial activity was observed against *S. pneumoniae*, with MIC values ranging from 1.3 to 2.6 µM for compounds **1–3**, **5**, and **6**. Similarly, compounds **2**, **5–7**, and **20–22** had MIC values ranging from 2.5 to 5 µM against *S. aureus*. The tested diruthenium compounds have a bactericidal effect significantly faster than that of penicillin. Fluorescence microscopy assays performed on *S. aureus* using the BODIPY-tagged diruthenium complex **15** showed that this type of metal compound enters the bacteria and does not accumulate in the cell wall of gram-positive bacteria. Cellular internalization was further confirmed by inductively coupled plasma mass spectrometry experiments. The nature of the substituents anchored on the bridging thiols and the compounds molecular weight appears to significantly influence the antibacterial activity. Thus, if overall a decrease of the bactericidal effect with the increase of compounds' molecular weight is observed, however, the complexes bearing larger benzo-fused lactam substituents had low MIC values. This first antibacterial activity screening demonstrated that the thiolato-diruthenium compounds exhibit promising activity against *S. aureus* and *S. pneumoniae* and deserve to be considered for further studies.

**IMPORTANCE** The *in vitro* assessment of diruthenium(II)-arene compounds against *Escherichia coli*, *Streptococcus pneumoniae*, and *Staphylococcus aureus* showed a significant antibacterial activity of some compounds against *S. pneumoniae*, with minimum inhibitory concentration (MIC) values ranging from 1.3 to 2.6 µM, and a medium activity against *E. coli*, with MIC of 25 µM. The nature of the substituents anchored on the bridging thiols and the compounds molecular weight appear to significantly influence the antibacterial activity. Fluorescence microscopy showed that these ruthenium compounds enter the bacteria and do not accumulate in the cell wall of gram-positive bacteria. These diruthenium(II)-arene compounds exhibit promising activity against *S. aureus* and *S. pneumoniae* and deserve to be considered for further studies, especially the compounds bearing larger benzo-fused lactam substituents.

**KEYWORDS** ruthenium complexes, *Escherichia coli*, *Streptococcus pneumoniae*, *Staphylococcus aureus*, MIC, fluorescence, uptake, ICP-MS, benzo-fused lactams

The increasing occurrence of multidrug-resistant (MDR) bacteria and difficult-to-treat infections associated with high morbidity and mortality have become important medical issues (1–3). Yearly, the number of estimated deaths caused by

Address correspondence to Markus Hilty, markus.hilty@unibe.ch, or Julien Furrer, julien.furrer@unibe.ch.

Markus Hilty and Julien Furrer are joint senior authors.

The authors declare no conflict of interest.

See the funding table on p. 16.

antibiotic-resistant infections is 33,000 in Europe and 35,000 in the United States (4). In 2019, on a global scale, 1.27 million deaths were directly attributable to drug resistance (5). Rapid and efficient solutions are required to counter potential outbreaks, with terrible life costs and economical damages (1, 6). A list of the world's leading antibiotic-resistant bacteria with an urgent demand for new antibiotics was published by World Health Organization (WHO) (7), aiming not only to favor the surveillance and control of MDR bacteria but also to favor research of new active compounds. Antibiotic-resistant *Enterobacteriaceae* (encompassing *Escherichia coli*), *Staphylococcus aureus* as well as *Streptococcus pneumoniae* are included in the WHO's list (7).

To efficiently face the problem of bacterial resistance to antibiotics [intrinsic and/or acquired (8)], the improvement of already existing antibiotics should be paralleled by the development of new classes of active compounds against which bacteria are less prone to develop resistances (9).

Metal complexes reached clinical trials for the treatment of various diseases and some compounds show also interesting antibacterial activity (10–12). However, the use of metal-based compounds as antibiotics is not widespread, and only a limited number of compounds have undergone clinical trials (13). Examples of metal complexes active against Gram-negative and Gram-positive bacteria (*E. coli* and *S. aureus*) are provided in Table S1 (13–32). Ruthenium is one of the metals considered for the development of metal-based antibiotics being generally associated to reduced toxicity (10). Some ruthenium complexes exert good activity against Gram-positive bacteria and low activity toward Gram-negative bacteria, notable exceptions being dinuclear polypyridylruthenium(II) complexes (31, 33) and ruthenium-based carbon-monoxide-releasing molecules (34–36).

Di- and trithiolato-bridged dinuclear ruthenium(II)-arene complexes have been investigated for more than a decade. This type of diruthenium complex is generally stable (37, 38) and showed promising *in vitro* and *in vivo* anticancer activity (37, 39) and *in vitro* antiparasitic activity against *Toxoplasma gondii* (40–44) and *Trypanosoma brucei* (45). Their exact mode of action has not yet been elucidated, but a profound alteration of the structure and activity of mitochondria has been identified (40, 41, 45).

To identify further biological applications of dinuclear thiolato-bridged ruthenium(II)-arene complexes, a library of 22 compounds was screened for potential antibacterial properties. This library included both previously reported and newly synthesized compounds with significant structural diversity: dithiolato compounds, symmetric and mixed trithiolato compounds, and conjugates containing one or more carbohydrate units, short lipid chains, benzo-fused lactams, and fluorophores. The minimum inhibitory concentration (MIC) values of the compounds were measured against various strains of clinically relevant Gram-negative (*E. coli*) and Gram-positive bacteria (*S. pneumoniae* and *S. aureus*). Additional experiments to evaluate the bacteriostatic and bactericidal properties of the compounds as well as their internalization and cellular localization were also performed.

## RESULTS

### Chemistry

Based on their structural features, the compounds included in this study were organized into four families.

### *Di- and trithiolato diruthenium compounds 1–8 (Family 1)*

Family 1 comprises eight previously known diruthenium complexes (**1–8**, Fig. 1) and includes the neutral dithiolato compound **4**, two symmetric trithiolato diruthenium complexes **5** and **6** with benzyl and phenyl substituents on the bridging thiols, and five mixed trithiolato diruthenium complexes **1–3**, **7,** and **8** in which various structural elements were varied. Some of these compounds have previously shown high anticancer and antiparasitic activity (44, 46, 47). For example, compound **1** exhibited nanomolar

**FIG 1** Structure of compounds **1–8** forming Family 1.

range IC$_{50}$ (half maximal inhibitory concentration) values against various parasites as *Neospora caninum* (41) and *T. gondii* (40).

## Compounds 9–12—diruthenium complexes conjugated to hexanoic acid and to carbohydrates (Family 2)

In this family, four trithiolato diruthenium compounds with organic molecules anchored on one of the bridge thiols were also selected. Compounds **9** and **10** (48) contain a hexanoic acid residue connected *via* an ester and, respectively, an amide bond to the diruthenium unit, and they were chosen considering that the lipophilic chain could influence membrane transfer (Fig. 2).

Compounds **11** and **12** present acetyl-protected D-glucose and D-galactose units, respectively (Fig. 2), anchored to one of the bridge thiols *via* a triazole linker, and they were selected in view of a potentially facilitated bacteria internalization due to the presence of the carbohydrate unit. The synthesis, characterization, and *T. gondii* antiparasitic effects of compounds **9–12** have been previously described (48, 49).

## Compounds 13–16—diruthenium complexes conjugated to BODIPY fluorophores (Family 3)

The development of fluorophore-labeled conjugates of metal-based drugs as traceable therapeutic agents has become extremely popular as they can provide important information relative to compounds cellular uptake, localization, and specific accumulation (42, 50). BODIPY (boron dipyrromethene, 4,4-difluoro-4-bora-3*a*,4*a*-diaza-*s*-indacene) is among the most attractive fluorophores tagged as they are generally non-toxic, photostable, and exhibit high fluorescence quantum yields (50).

**FIG 2** Structure of diruthenium conjugates **9–12** forming Family 2.

Four fluorescent conjugates, **13–16**, containing a BODIPY dye unit attached to the diruthenium moiety through linkers of various lengths *via* ester (compounds **13** and **15**) or amide (compounds **14** and **16**) bonds (Fig. 3), were selected. Although an important fluorescence quenching was observed after conjugating the BODIPY to the diruthenium unit, compounds **13–16** could be used as fluorescent tracers (51) and were recently investigated as potential anti-toxoplasma agents. Fluorescence microscopy investigations of these compounds showed a pattern of cytoplasmic, but not nuclear, localization in human foreskin fibroblasts (HFFs) (51).

### Compounds 17–22—diruthenium compounds with various thiol ligands (ortho-substituted arenes and benzo-fused lactams as pending groups; Family 4)

Six new thiolato diruthenium compounds **17–22** (Fig. 4) were synthesized to evaluate structurally different ligands. The description of the synthetic protocols and the analysis and characterization of these compounds and the corresponding ligands are provided in the *Supporting information* [Fig. 5 and 6 for the synthesis of the diruthenium complexes, and Schemes S1 and S2 (*Supporting information*) for the synthesis of the thiol ligands].

In compounds **17–19** (Fig. 5), three different groups, diphenylphosphonate (17), 2–4,4,5,5-tetramethyl-1,3,2-dioxaborolan-2-yl (18), and aldehyde (19) were introduced in the *ortho* position rather than in *para* position of one of the bridging thiol-ligand. The synthesis of the ligands **17**a [diphenyl-(2-mercaptophenyl)phosphonate], **18**a [2-(4,4,5,5-tetramethyl-1,3,2-dioxaborolan-2-yl)benzenethiol], and **19**a (2-mercaptobenzaldehyde; Schemes S1) is presented in detail in the Supporting information.

For compounds **20–22** (Fig. 5), two benzo-fused lactams containing thiol groups [6-mercapto-3,4-dihydroquinolin-2(1*H*)-one (**22** a) and 7-mercapto-1,3,4,5-tetra-hydro-2*H*-benzo(*b*)azepin-2-one (**23**)] were used as bridging ligands instead of 4-mercap-tophenol in the parent trithiolato diruthenium compound **1**. Their synthesis is provided

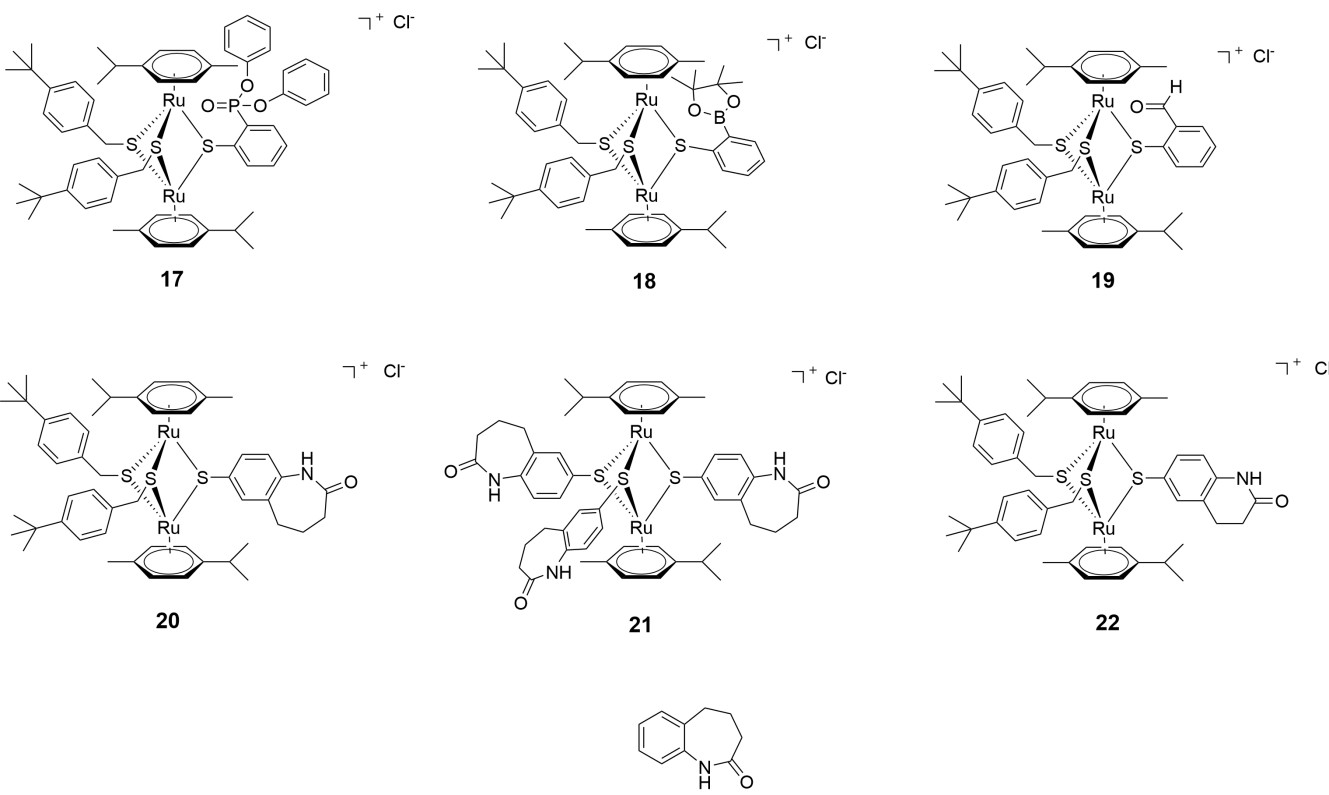

**FIG 3** Structure of diruthenium-BODIPY conjugates **13–16** forming Family 3.

in detail in the *Supporting information* and were selected because benzo-fused lactams exhibit various biological activities among which antimicrobial properties (52, 53).

**FIG 4** Structure of compounds **17–22** forming Family 4. **23** is the benzo-fused lactam thiol ligand used in compounds **20** and **21** and tested against various bacteria.

**FIG 5** Synthesis of mixed trithiolato diruthenium complexes **17–20** and **22**.

In the symmetric trithiolato diruthenium complex **21** (Fig. 6), the benzyl and phenyl bridging thiols in complexes **5** and **6** were replaced by three 7-mercapto-1,3,4,5-tetrahydro-2H-benzo[b]azepin-2-one units.

Asymmetric trithiolato diruthenium complexes **17–20** and **22** were synthetized by reacting the dithiolato intermediate **4** with an excess of the corresponding ligands **17a–19a**, **22a**, and **23** (in 1,2-dichloroethane -DCE- at 100°C or in dichloromethane at 60°C). In the case of complexes **17–19**, excess of triethylamine was employed to ensure the completion of the reaction. All the compounds were isolated in moderate yields 41%–65% (Fig. 5).

The symmetric trithiolato diruthenium complex **21** was obtained by reacting the commercially available ruthenium dimer {[Ru($\eta^6$-p-MeC$_6$H$_4$Pr$^i$)Cl]$_2$Cl$_2$} with an excess of

**FIG 6** Synthesis of the symmetric trithiolato diruthenium complex **21**.

the thiol ligand **23** in refluxing EtOH in basic conditions ($K_2CO_3$) and was isolated in 56% yield (Fig. 6).

## Stability of the diruthenium compounds

For the biological activity evaluation, 1 mM stock solutions of all compounds were prepared in dimethylsulfoxide (DMSO). $^1$H-NMR spectra of several conjugates dissolved in DMSO-$d_6$, recorded at 25°C 5 min, and 100 days after sample preparation showed no modifications, demonstrating very good stability of the diruthenium compounds in this highly complexing solvent (42, 44, 51).

Compounds **9**, **11–13**, and **15** present a carboxyl ester bond that can potentially be hydrolyzed in cellular growth media. Compounds **13** and **15** as well as other similar conjugates with coumarin and BODIPY fluorescent units have been recently studied (42, 51), and for these compounds, fluorescence measurements proved that the ester bonds remained intact for 48 h, but very limited hydrolysis of the ester bonds with the release of coumarin and BODIPY dyes after 168 h could be observed. It was concluded that diruthenium conjugates with ester linkers were stable under the conditions used for biological evaluations, and therefore, it was assumed that compounds **9**, **11–13**, and **15** are sufficiently stable during the *in vitro* evaluation.

## Determination of the MIC values on *E. coli*

The MIC values (lowest concentration of antibiotic at which bacterial growth is completely inhibited) of the compounds were determined on three *E. coli* strains [P53.1R (54), P54.1T (54), and P54.2R (54)], the *S. pneumoniae* D39 (NCTC 7466) (55) strain, and the *S. aureus* 20 (54) strain. Of note, the P53.1R and P54.2R strains were extended-spectrum beta-lactamases resistant, while the P54.1T strain was colistin resistant. A table (Table S1) summarizing the nature and resistance profiles of the three strains is included in the Supplementary Information.

The first screening of the 22 diruthenium compounds was conducted on *E. coli* P53.1R, a colistin sensitive but extended spectrum cephalosporin resistant strain (Table 1). Most of the compounds exhibited no or very low antibacterial activity on this Gram-negative bacterium [MIC values over or at the limit of the concentration range (0.01–100 µM) used in the assays]. From this library, the only compound inhibiting *E. coli* P53.1R cells growth with medium activity (MIC value of 25 µM) was the symmetric diruthenium compound **5** (Table 1). The potency of compounds **2**, **5**, and **6** from Family 1 was further assessed for two additional *E. coli* strains P54.1T and P54.2R (Table 1). On the *E. coli* P54.1T and P54.2R strains, the lowest MIC values (of 25 and 20.8 µM, respectively) were obtained for the same compound **5** (Table 1), which inhibited the bacteria growth with medium efficiency compared with the reference drug colistin for which the MIC value is 0.8 µM.

## Determination of the MIC values on *S. pneumoniae*

In a second screening assay, the MIC values of diruthenium complexes **1–12** on *S. pneumoniae* D39 strain were determined, proving to be significantly lower compared to those measured on the three *E. coli* strains (Table 1). Compounds **5** and **6** were most effective (MIC value of 1.3 µM for both compounds) and slightly more active than compounds **1–3** (MIC values of 1.8–2.6 µM). The MIC values corresponding to compounds **7** and **8** were at the limit of the concentration used for the assay (0.625–10 µM; Table 1) and being more than five times higher compared to the MIC value of compounds **5** and **6**. From Family 1, the dithiolato derivative **4** exhibited the lowest activity on *S. pneumoniae* D39. The MIC values of compounds **9–12** (Family 2) on *S. pneumoniae* D39 were higher than those found trithiolato compounds **1–3** and **5–7**.

**TABLE 1** MIC values measured for the diruthenium compounds **1–22**, the ligand **23**, and colistin and penicillin used as controls[c]

| | MIC (µM and µg/mL) | | | | |
|---|---|---|---|---|---|
| | *E. coli* | | | *S. pneumoniae* | |
| Compound | P53.1R[a] | P54.1T[a] | P54.2R[a] | D39 (NCTC 7466)[b] | *S. aureus* 20 [a] |
| Family 1 | | | | | |
| 1 | >80.0 | n.d. | n.d. | 2.5 ± 1.0 | 17.5 ± 5.0 |
| | | | | (2.5 ± 1.0) | (17.3 ± 4.9) |
| 2 | 100.0 ± 0.0 | 100.0 ± 0.0 | 100.0 ± 0.0 | 1.8 ± 0.5 | 12.5 ± 5.0 |
| | (98.9 ± 0.0) | (98.9 ± 0.0) | (98.9 ± 0.0) | (1.8 ± 0.5) | (12.4 ± 5.0) |
| 3 | >80.0 | n.d. | n.d. | 2.6 ± 0.8 | 5 ± 0.0 |
| | (>82.5) | | | (2.7 ± 0.8) | (5.2 ± 0.0) |
| 4 | >80.0 | n.d. | n.d. | 17.5 ± 5.0 | 12.5 ± 5.0 |
| | (>72.0) | | | (15.8 ± 4.5) | (11.3 ± 4.5) |
| 5 | **25.0 ± 0.0** | **25.0 ± 0.0** | **20.8 ± 7.2** | **1.3 ± 0.6** | 4.4 ± 1.3 |
| | (21.9 ± 0.0) | (21.9 ± 0.0) | (18.2 ± 6.3) | (1.1 ± 0.5) | (3.9 ± 1.2) |
| 6 | 100.0 ± 0.0 | 100.0 ± 0.0 | 66.7 ± 28.9 | **1.3 ± 0.6** | **2.5 ± 0.0** |
| | (83.4 ± 0.0) | (83.4 ± 0.0) | (55.6 ± 24) | (1.1 ± 0.5) | (2.1 ± 0.0) |
| 7 | >80.0 | n.d. | n.d. | 6.7 ± 2.9 | 5.0 ± 0.0 |
| | (>76.9) | | | (6.4 ± 2.8) | (4.8 ± 0.0) |
| 8 | >80.0 | n.d. | n.d. | 10.0 ± 0.0 | 6.7 ± 2.9 |
| | (>83.2) | | | (10.4 ± 0.0) | (7.0 ± 3.0) |
| Family 2 | | | | | |
| 9 | >80.0 | n.d. | n.d. | 10.0 ± 0.0 | 40.0 ± 0.0 |
| | (>87.0) | | | (10.9 ± 0.0) | (43.5 ± 0.0) |
| 10 | >80.0 | n.d. | n.d. | > 10.0 | 80.0 ± 0.0 |
| | (>87.0) | | | (>10.9) | (87.0 ± 0.0) |
| 11 | 80.0 ± 0.0 | n.d. | n.d. | 10.0 ± 0.0 | 10.0 ± 0.0 |
| | (115.5 ± 0.0) | | | (10.4 ± 0.0) | (14.4 ± 0.0) |
| 12 | 80.0 ± 0.0 | n.d. | n.d. | 10.0 ± 0.0 | 8.3 ± 2.9 |
| | (115.5 ± 0.0) | | | (10.4 ± 0.0) | (12.0 ± 4.2) |
| Family 3 | | | | | |
| 13 | >80.0 | n.d. | n.d. | n.d. | 80.0 ± 0.0 |
| | (>112.3) | | | | (112.3 ± 0.0) |
| 14 | >80.0 | n.d. | n.d. | n.d. | >80.0 |
| | (>115.7) | | | | (>115.7) |
| 15 | >80.0 | n.d. | n.d. | n.d. | 80.0 ± 0.0 |
| | (>114.6) | | | | (114.6 ± 0.0) |
| 16 | >80.0 | n.d. | n.d. | n.d. | >80.0 |
| | (>108.9) | | | | (>108.9) |
| Family 4 | | | | | |
| 17 | >80.0 | n.d. | n.d. | n.d. | 40.0 ± 0.0 |
| | (>96.5) | | | | (48.2 ± 0.0) |
| 18 | >80.0 | n.d. | n.d. | n.d. | 8.8 ± 2.5 |
| | (>88.0) | | | | (9.7) |
| 19 | >80.0 | n.d. | n.d. | n.d. | 15.0 ± 5.8 |
| | (>80.1) | | | | (15.0 ± 5.8) |
| 20 | 100.0 ± 40 | n.d. | n.d. | n.d. | 3.8 ± 1.4 |
| | (105.7) | | | | (4.0 ± 1.5) |
| 21 | >80.0 | n.d. | n.d. | n.d. | **2.5 ± 0.0** |
| | (>86.6) | | | | (2.7 ± 0.0) |
| 22 | >80.0 | n.d. | n.d. | n.d. | **3.1 ± 1.3** |
| | (>83.4) | | | | (3.2 ± 1.3) |

*(Continued on next page)*

**TABLE 1** MIC values measured for the diruthenium compounds **1–22**, the ligand **23**, and colistin and penicillin used as controls[c] (*Continued*)

| | MIC (µM and µg/mL) | | | | |
|---|---|---|---|---|---|
| | *E. coli* | | | *S. pneumoniae* | |
| Compound | P53.1R[a] | P54.1T[a] | P54.2R[a] | D39 (NCTC 7466)[b] | *S. aureus* 20 [a] |
| Ligand and reference drugs | | | | | |
| 23 | >240.0 | n.d. | n.d. | n.d. | >30.0 |
| | (>46.4) | | | | (>5.8) |
| Colistin | 0.8 ± 0.6 | 5.8 ± 2.0 | 1.1 ± 2.0 | n.d. | n.d. |
| | (0.9 ± 0.5) | (6.7 ± 2.3) | (1.3 ± 2.4) | | |
| Penicillin | n.d. | n.d. | n.d. | <0.02 | 0.09 ± 0.03 |
| | | | | (0.01) | (0.03 ± 0.01) |

[a]Information in reference (54).
[b]Information in reference (55).
[c]The reported values are the means of three biological replicates ± SD. "n.d." stands for "not determined." The lowest MIC values for each strain are highlighted in bold. The values in brackets are the values expressed in µg/mL

## Determination of the MIC values on *S. aureus*

In the third screening, MIC values of the 22 compounds on *S. aureus* 20 strain were also evaluated (Table 1). In Family 1, compound **6**, with an MIC value of 2.5 µM, was the most potent, followed by compounds **3**, **5**, and **7** (MIC values ranging from 4.4 to 5 µM). Hydroxy compound **1** was the least efficient (MIC values of 17.5 µM), followed by the amino derivative **2**, and the dithiolato compound **4** (MIC value of 12.5 µM for both). Interestingly, compared to **1**, slight structural modifications in compound **7** ($^i$Pr group instead of $^t$Bu as substituents on two of the bridging thiols) lead to an improvement of activity against *S. aureus* 20 (MIC values of 17.5 and 5.0 µM for **1** and **7**, respectively). A similar effect was observed in the case of compound **8** compared to **2** for which the presence of $BF_4^-$ as counterion instead of $Cl^-$ was associated to a decrease of the MIC value on *S. aureus* 20 from 12.5 to 6.7 µM for **2** and **8**, respectively.

Interestingly, for compounds **1** and **2**, important differences of activity are observed on the two tested Gram-positive bacteria *S. aureus* 20 and *S. pneumoniae* D39.

In Family 2, conjugates **9** and **10**, with a hexanoic acid residue attached on one of the bridging thiols *via* ester or amide bonds, showed poor activity on *S. aureus* 20 (MIC values of 40 and 80 µM, respectively). Interestingly, the acetyl-protected glucose and galactose hybrids **11** and **12** exhibited lower MIC values (11 µM and 8.3 µM, respectively) but are up to four times less efficient in inhibiting the bacteria growth compared to **6**.

All the diruthenium compounds conjugated to BODIPY fluorophores (Family 3, compounds **13–16**) exhibited low activity against *S. aureus* 20 (MIC values over 80 µM).

Contrasting MIC values were obtained for compounds **17–22** (Family 4). From compounds **17–19**, bearing *ortho*-substituted ligands, derivative **17** with a diphenyl-(2-mercaptophenyl)phosphonate bridging ligand, exhibited reduced efficacy on *S. aureus* 20 (MIC value of 40 µM), while the boron-containing compound **18** and aldehyde compound **19** exhibited significantly lower MIC values (8.8 µM and 15 µM, respectively). Compounds **20–22**, bearing at least one benzo-fused lactam unit on the bridging thiols, exhibited low MIC values of 3.8, 2.5, and 3.1 mM, respectively.

The antibacterial activity of ligand **23** [7-mercapto-1,3,4,5-tetrahydro-2*H*-benzo(*b*)azepin-2-one] was also evaluated on *S. aureus*, using three times the concentration employed in the tests with compound **21**, but no bacteria growth inhibition was measured within the concentration range of the assay (7.5–240 µM).

The average MIC values of the 22 diruthenium compounds on the tested bacteria as a function of their respective molecular weight are shown in Fig. S10, as well as the linear regression for each bacterial species. Interestingly, Fig. S10 suggests a slight inverse correlation between MIC values and molecular weight for all bacterial species investigated in the present study. While this trend hardly exists for the *E. coli* phenotype P53.1R (only five compounds are considered), it appears evident for *S. pneumoniae* D39 and especially for *S. aureus* 20).

## Toxicity

The toxicity of compounds **1–16** against HFFs has been previously evaluated (44, 48, 49, 56). The data have been compiled in Table S2. Briefly, compounds **3**, **5**, **6**, **12**, **13**, **15**, and **16** were classified as very promising, with no toxic effect on HFF cells at a high concentration of 2.5 µM, (survival rate of HFF >90%) compounds **1**, **2**, **7**, and **8** were moderately toxic (survival rate >50%), and compounds **9** and **10** were very toxic on HFF (survival rate 0%).

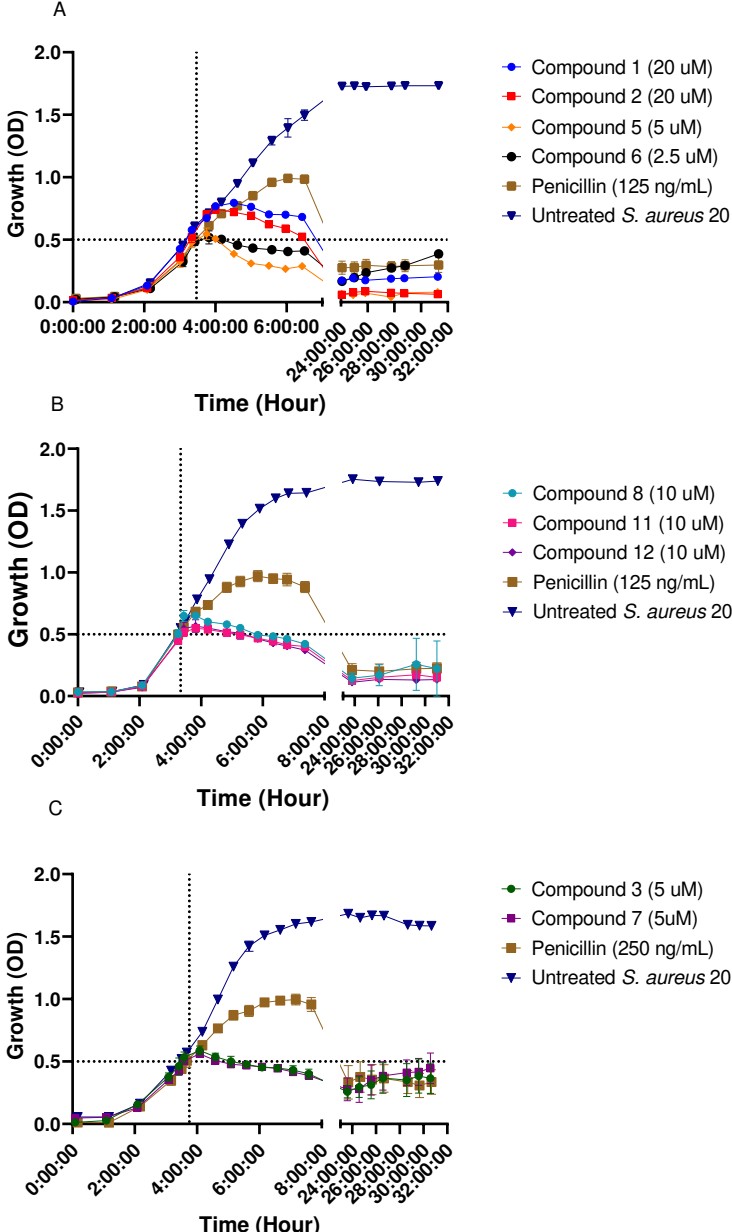

**FIG 7** Growth curves of *S. aureus* 20 after the addition of the diruthenium compounds at MIC when $OD_{600}$ = 0.5 is reached (dotted lines). (A) *S. aureus* treated with compounds **1–2** and **5–6** at MIC upon reaching $OD_{600}$ = 0.5. (B) *S. aureus* treated with compounds **8**, **11**, and **12** at MIC upon reaching $OD_{600}$ = 0.5. (C) *S. aureus* treated with compounds **3** and **7** at MIC upon reaching $OD_{600}$ = 0.5. Treatment with penicillin at four times its MIC was used as a bactericidal control.

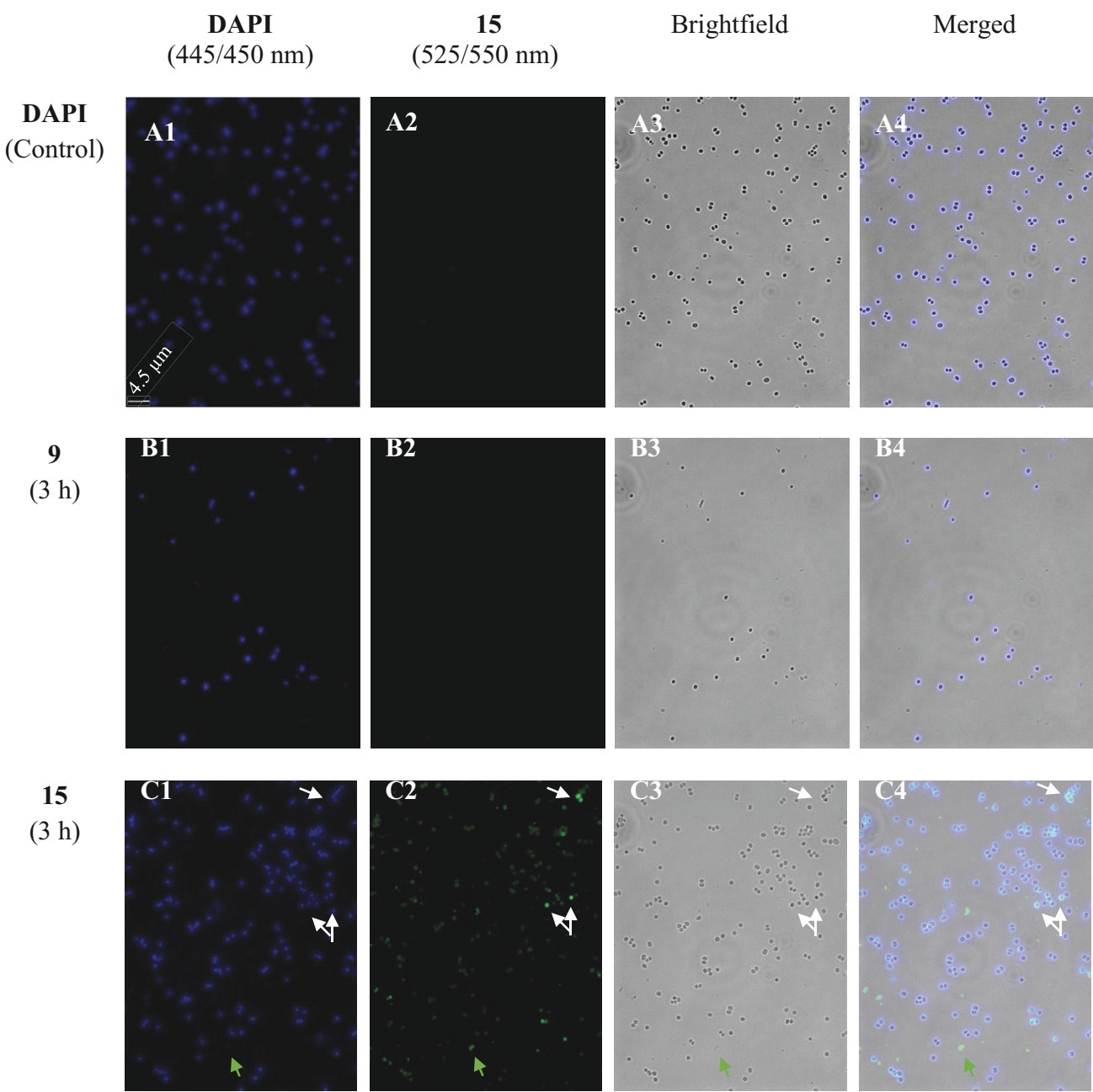

**FIG 8** Fluorescence microscopy images of *S. aureus* 20 cells stained with DAPI after 3 h of incubation, with or without treatment with diruthenium conjugates **9** and **15**. (A) Untreated control sample stained with DAPI. (B) The sample was treated with **9** for 3 h and stained with DAPI for visualization. (C) The sample was treated with **15** for 3 h and stained with DAPI for visualization. **15** is visible after excitation at 440/470 nm with emission at 525/550 nm. White arrows indicate cells with overlapped fluorescence of DAPI and BODIPY diruthenium conjugate **15**. Green arrows indicate emission in 525/550 nm (corresponding to conjugate **15**) that do not overlap with DAPI nor were cells visible under Brightfield.

## Evaluation of the bacteriostatic and bactericidal properties

The bacteriostatic and bactericidal properties of selected diruthenium compounds **1–3**, **5–8**, and **11–12** were investigated on *S. aureus* 20 (Fig. 7). Interestingly, the compounds exhibited a significantly faster effect compared to penicillin, even when the penicillin concentration was increased to eight times its MIC value (250 ng/L). The sharp decrease

of the optical density at 600 nm ($OD_{600}$) after the addition of all diruthenium compounds and the complete absence of cell growth within 24 h suggests a bactericidal rather than bacteriostatic effect.

## Fluorescence microscopy

To determine whether the compounds preferentially accumulate inside the bacteria or on their membrane, conjugate **15** bearing a fluorescent BODIPY tag was used to investigate the localization of the diruthenium compounds by fluorescence microscopy. The hexanoic ester conjugate **9** was also tested in the fluorescence microscopy experiments as a negative control (the fluorescence of **9** at 525/550 nm is shown in Fig. 8, panels B). In parallel with compounds **9** and **15**, DAPI (4,6-diamidin-2-phenylindol) stain was used for the visualization of nuclear DNA.

In Fig. 8 (panels C), an overlap of the blue fluorescent DAPI marker and the green fluorescence of BODIPY conjugate **15** inside the cells is clearly visible. Interestingly, emissions from **15** which do not overlap with those of DAPI are also observed (Fig. 8, panels C). The results only validate the nuclear DNA as intracellular target of DAPI. The relative low fluorescence signal of compound **15** (51) does not allow to clearly distinguish the exact intracellular compound localization which might be the cytoplasm or the inner/outer membrane. Additional images are presented in Fig. S12 to S15 in *Supporting information*.

## ICP-MS and cell counting

The cellular uptake of compounds **5**, **6**, **9**, **15**, and **21** in the *S. aureus* 20 strain was quantified using inductively coupled plasma mass spectrometry (ICP-MS) measurements, the results being summarized in Table 2. The bacteria were incubated with the selected ruthenium compounds at their respective MIC value for 1, 2, and 3 h. A clear increase of the ruthenium content of treated *S. aureus* 20 cells compared to the untreated control [mean value $(6.6 \pm 2.2) \times 10^{-17}$ µM/CFU(colony-forming unit)] was measured, indicating cellular uptake of the diruthenium compounds (Table 2).

The amount of compound internalized by the cells appears to be independent of the treatment duration (Table 2). If overall, within the treatment time frame of 3 h, no significant time-dependent modifications were observed for the same compound, important variations between the tested compounds were noticed. For example, the measured ruthenium concentration in *S. aureus* 20 cells after treatment with compound

**TABLE 2** Content of ruthenium in *S. aureus* 20 cells measured by ICP-MS[a]

| Compound | Incubation time (h) | Average compound concentration *per S. aureus* cells (µM/CFU) |
|---|---|---|
| 5 | 1 | $(7.7 \pm 2.0) \times 10^{-13}$ |
| | 2 | $(4.2 \pm 1.5) \times 10^{-13}$ |
| | 3 | $(4.1 \pm 1.3) \times 10^{-13}$ |
| 6 | 1 | $(4.0 \pm 1.2) \times 10^{-13}$ |
| | 2 | $(1.4 \pm 1.4) \times 10^{-12}$ |
| | 3 | $(1.4 \pm 1.1) \times 10^{-12}$ |
| 9 | 1 | $(2.8 \pm 1.9) \times 10^{-13}$ |
| | 2 | $(2.4 \pm 1.8) \times 10^{-13}$ |
| | 3 | $(2.6 \pm 1.3) \times 10^{-13}$ |
| 15 | 1 | $(2.2 \pm 1.7) \times 10^{-13}$ |
| | 2 | $(3.6 \pm 2.7) \times 10^{-13}$ |
| | 3 | $(3.4 \pm 2.6) \times 10^{-13}$ |
| 21 | 1 | $(6.6 \pm 0.6) \times 10^{-14}$ |
| | 2 | $(5.9 \pm 0.3) \times 10^{-14}$ |
| | 3 | $(6.1 \pm 1.2) \times 10^{-14}$ |

[a]Values are the means of the three replicates ± SD.

**6** ($1.4 \times 10^{-12}$ µM/CFU) was more than 20 times higher compared with the value for compound **21** ($6 \times 10^{-14}$ µM/CFU).

The results suggest a possible correlation between the amount of diruthenium compound internalized by the bacteria and the molecular weight, but no dependence on the respective MIC values (Fig. S11). Compound **21**, bearing three 7-mercapto-1,3,4,5-tetrahydro-2*H*-benzo(*b*)azepin-2-one ligands as bridging thiols, is particularly interesting. **21** exhibits a low MIC value (2.5 µM, Table 1) but less efficient cellular uptake. These data suggest that **21** is highly toxic to bacteria, and this compound can be considered as a lead for further compound development through appropriate chemical modifications.

## DISCUSSION

In this work, the *in vitro* properties of a library of 22 trithiolato-bridged dinuclear ruthenium(II)-arene complexes as potential antibacterial agents against relevant bacterial species were investigated. The *in vitro* antiparasitic and anticancer activity of compounds **1–16** have been previously studied (40, 41, 44, 57). This type of diruthenium complex is known to target in the mitochondria (40, 41, 44, 45, 58) an organelle sharing many features with bacterial cells (59).

Following the trend of most antibiotics [being more effective against Gram-positive than on Gram-negative pathogens (8, 12)], the results of this study revealed that the MIC values of the 22 tested compounds were higher on *E. coli* strains than on *S. pneumoniae* and *S. aureus*. Moreover, compared to colistin and penicillin tested as reference drugs, even the most effective compounds in this library were only moderately active, showing MIC values more than 10 times higher.

For comparison, the MIC values of the ruthenium compounds *cis*-α-[Ru(phen)bb$_{12}$]$^{2+}$ and *cis*-β-[Ru(phen)bb$_{12}$]$^{2+}$ against the *S. aureus* MSSA ATCC 25,923 strain were 0.5 µM, and against the *E. coli* ATCC 25922 strain they were 8.3 and 16 µM, respectively (24). On the other hand, the MIC values of the ruthenium compound [Ru(bpy)$_2$(methionine)]$^{2+}$ against the *S. aureus* MSSA ATCC 25,923 strain and against the *E. coli* ATCC 11303 strain were of 73.3 and 586.4 µM, respectively (21).

The physico-chemical properties of the bridging thiol ligands are very important, their influence appearing particularly important in compounds **20–22** bearing benzo-fused lactams substituents, which exhibited low MIC values, and to a lesser extent, for carbohydrate conjugates **11–12**. The neutral dithiolato complex **4** had medium MIC values against *E. coli* and *S. pneumoniae*, following the trend observed with other dithiolato complexes against cancer cells and parasites (40, 47, 57). The lack of activity against cancer cells and parasites was attributed to the lability of the chloride ligands and the general lower stability of dithiolato compounds (60, 61). Yet, the MIC value of **4** against *S. aureus* was relatively low and comparable to that of the mixed trithiolato compounds **1** and **2**, compounds that were otherwise much more active than dithiolato complexes against cancer cells and protozoan parasites (40, 62). This difference is appealing and needs further investigations.

The cellular uptake of the BODIPY-diruthenium conjugate **15** by *S. aureus* was confirmed by fluorescence microscopy investigations. The images presented in Fig. S15 (*Supporting information*) indicate that compound **15** does not have a specific cellular localization. Fig. S15 clearly shows that compound **15** does not accumulate in membranes or cell walls but concentrates inside cells, forming aggregates that potentially lead to cell death.

In conclusion, the results show that the lower molecular weight diruthenium compounds tend to have lower MIC values against Gram-positive bacteria, but the presence of specific substituents, especially the 1,3,4,5-tetrahydro-2*H*-benzo(*b*)azepin-2-one unit, on the bridge thiols strongly influence the antibacterial activity. The 7-mercapto-1,3,4,5-tetrahydro-2*H*-benzo(*b*)azepin-2-one is an interesting ligand to be considered for the development of other potent antibacterial metal complexes. Finally, the localization and the cellular uptake of the compounds assessed using fluorescence

microscopy and ICP-MS, respectively, suggest that the cellular walls as well as the nucleic acid or protein synthesis are not the main targets of this type of compounds.

## MATERIALS AND METHODS

### Materials

All new diruthenium complexes were synthesized, analyzed, and characterized at the Department of Chemistry, Biochemistry, and Pharmaceutical Sciences of University of Bern according to the methods provided in the *Supporting information*. Colistin, penicillin G potassium salt, and gentamicin sulfate were purchased from Sigma-Aldrich and used as received.

### Instrumentation

The $OD_{600}$ of cell cultures and of the tubes used for the assays were measured with a Helios Epsilon spectrophotometer. In the MIC assays, the absorbance values for wells plates were obtained using a Varioskan Flash spectral scanning multimode plate reader using the SkanIt software. Cell lysis was realized using a Branson Ultrasonics Sonifier 250. Fluorescence microscopy observations were conducted using a ZEISS AXIO Imager M1, fluorescence microscope. ICP-MS values were determined with a NexION 2000B ICP Mass Spectrometer using Ir-193 as internal standard.

### Antibacterial activity

#### Cell culture

In this work were used three extended-spectrum cephalosporin-resistant *E. coli* strains (P53.1R, P54.1T, and P54.2R) with varied colistin sensitivity, a penicillin-sensitive strain of *S. pneumoniae* (D39) and a penicillin-sensitive strain of *S. aureus* (20). The *E. coli* and *S. aureus* strains were inoculated in Luria-Bertani agar plates and incubated overnight at 37°C. *S. pneumoniae* was inoculated in Columbia blood sheep agar (CSBA) plates and incubated overnight at 37°C and 5% $CO_2$. Individual colonies of *E. coli* were then subcultured overnight at 37°C under constant agitation in Luria-Bertani medium for initial screening and in Brain Heart Infusion (BHI) for the other assays. After the incubation period, the bacterial cultures were adjusted to a final concentration of $2 \times 10^7$ CFU/mL. Individual colonies of *S. pneumoniae* and *S. aureus* were subcultured in BHI at 37°C until they reached the appropriate concentrations of $1 \times 10^8$ and $2 \times 10^8$ CFU/mL, respectively. The final concentration for inoculation in the assays was $5 \times 10^5$, $5 \times 10^6$, and $1 \times 10^7$ CFU/mL, respectively.

#### Determination of the MIC

The assays were conducted in sterile Nunclon 96 microwell plates and Nunc 384 well polystyrene plates purchased from Thermo Fisher Scientific. DMSO stock solutions of the diruthenium complexes (1–5 mM) were diluted in $H_2O$ to concentrations ranging from 2 to 200 µM according to compounds' potency and dispensed into the plates with the previously adjusted bacterial suspension. The plates with *E. coli* or *S. aureus* were then incubated at 37°C for 18 h, and then the OD at 450 nm was measured using a Varioskan Flash, spectral scanning multimode plate reader. The plates with *S. pneumoniae* were incubated at 37°C for 22 h with agitation and measurement every 30 min. The MIC was determined as the lowest compound concentration showing a complete inhibition of visible bacterial growth. For comparison, all assays included antibiotic controls: colistin, 8 µg/mL (*E. coli* P53.1R and P54.2R), 64 µg/mL (*E. coli* P54.1T), and penicillin 0.5 µg/mL (*S. pneumoniae* and *S. aureus*), as well as DMSO controls to account for the lethality of the organic solvent.

## Determination of the bacteriostatic/bactericidal effects

The assay was conducted in test tubes read with a Helios Epsilon spectrophotometer. DMSO stock solutions of the diruthenium complexes were diluted in $H_2O$ to concentrations 50 times their MIC values, before being added into *S. aureus* bacterial suspension in the test tube at 1:50 of the total volume, thus reaching MIC. The tubes were incubated at 37°C in a water bath for 32 h. At the beginning of the assay, the OD of each sample was measured every hour until an $OD_{600} = 0.5$ was reached. At this point, the previously prepared solution of diruthenium complexes and antibiotics was quickly added to their respective tubes and the negative growth controls before continuing incubation. Subsequently, the OD was measured for all samples every half hour until a total incubation time of 8 h was reached. Between 8 and 24 h of incubation, the OD was not measured and then measured sporadically between 24 and 32 h of incubation to observe potential changes until the end of the assay.

## Cell counting

First, in a test tube, 5 mL BHI was added to 266.6 µL of a suspension of *S. aureus* in BHI and incubated at 37°C until an $OD_{600} = 0.5$ was measured. Subsequently, sequential dilutions were prepared, by adding 900 µL of a 0.85% NaCl aqueous solution to 100 µL from the suspension of *S. aureus* in BHI. This dilution procedure was repeated until a dilution of $10^{-9}$ was reached. 100 µL of the resulting $10^{-3}$ to $10^{-9}$ solutions were then put on CSBA plates and incubated at 37°C and 5% $CO_2$ for 18 h. The CFU/mL of the original suspension was then estimated by counting the number of colonies in each plate and using a calibration curve. Each measurement was realized in triplicate.

## Fluorescence microscopy

Samples were prepared by incubating *S. aureus* bacterial suspensions for 3 h at 37°C in a water bath. Then 5 µM of compounds **9** and **15** were added, and the suspensions were further incubated for 1, 2, and 3 h in 5 mL of BHI. Next, the samples were sedimented by centrifugation at 1,788 g for 12 min at 4°C and washed twice with 0.85% aqueous NaCl. Then, the pellets were re-suspended in 3 mL 0.85% aqueous NaCl and cooled on ice before adding 2 µL of DAPI [DAPI Solution (1 mg/mL), Thermo Scientific, 4′,6-diamidino-2-phenylindole, a fluorescent stain that binds strongly to adenine–thymine-rich DNA regions] stock solution. The samples were finally studied using a fluorescence microscope with filters for DAPI (excitation 365 nm and emission 445/50), GFP (green fluorescent compound, excitation 440/70, and emission 525/5) and brightfield. To ensure observation of the cell internal, Z-stacks pictures were taken with an interval of 320 nm, using an objective 100×, 0.065 µm/pixel.

## Inductively coupled plasma mass spectrometry

The samples were prepared by incubating *S. aureus* bacterial suspensions (~$4.3 \times 10^8$ CFU/mL) for 3 h at 37°C in a water bath. Then 1 mM solution of compounds **5**, **6**, **9**, **15**, and **21** in DMSO was added, and the suspensions were incubated for 1, 2, and 3 h. A 150 µg/mL penicillin solution in distilled water was used to ensure cell death. The samples were then sedimented by centrifugation at 1,788 g for 12 min at 4°C and washed with 0.85% aqueous NaCl twice. Afterward, the pellets were resuspended in 3 mL $H_2O$ and frozen at −20°C overnight before undergoing heat shock and sonication for cell lysis. The resulting samples were then filtered using 0.22 µm Millipore syringe filters and analyzed by ICP-MS.

## Determination of cellular uptake of the ruthenium compounds

Using the data obtained from the cell counting and ICP-MS experiments and assuming comparable losses during filtration for both samples and controls, the intracellular ruthenium content was calculated using Equation 1:

$$\mu\text{g Ru per cells} = \frac{\text{Sample concentration (ppb Ru} = \mu\text{g Ru/L)/Filtration losses (\%)}}{10 \text{ mL/3 mL} * \text{OD600} * 4.3 \cdot 10^8 \text{ CFU/ mL}} \quad (1)$$

A calibration curve using ruthenium solutions of 0.02 ppb, 0.1 ppb, 1 ppb, 5 ppb, and 50 ppb in 2% $HNO_3$ was first established, and Iridium 193 was used as internal standard.

## ACKNOWLEDGMENTS

This research was funded by the Swiss Nationals Science Foundation, Sinergia Project, grant number CRSII5-173718.

## AUTHOR AFFILIATIONS

[1]Department of Medicine, Institute for Infectious Diseases, University of Bern, Bern, Switzerland
[2]Department of Chemistry, Biochemistry and Pharmaceuticals Sciences, University of Bern, Bern, Switzerland

## AUTHOR ORCIDs

Markus Hilty  http://orcid.org/0000-0002-2418-6474
Julien Furrer  http://orcid.org/0000-0003-2096-0618

## FUNDING

| Funder | Grant(s) | Author(s) |
| --- | --- | --- |
| Schweizerischer Nationalfonds zur Förderung der Wissenschaftlichen Forschung (SNF) | CRSII5-173718 | Julien Furrer |

## AUTHOR CONTRIBUTIONS

Quentin Bugnon, Formal analysis, Investigation, Visualization, Writing – original draft | Camilo Melendez, Conceptualization, Formal analysis, Investigation, Methodology, Supervision, Visualization, Writing – original draft | Oksana Desiatkina, Formal analysis, Investigation, Writing – review and editing | Louis Fayolles de Chaptes, Formal analysis, Investigation | Isabelle Holzer, Formal analysis, Investigation | Emilia Păunescu, Conceptualization, Formal analysis, Investigation, Methodology, Supervision, Validation, Visualization, Writing – review and editing | Markus Hilty, Conceptualization, Methodology, Project administration, Resources, Supervision, Validation, Writing – original draft, Writing – review and editing | Julien Furrer, Conceptualization, Funding acquisition, Methodology, Project administration, Resources, Supervision, Validation, Writing – original draft, Writing – review and editing

## DATA AVAILABILITY

All the data are presented in this study and in the corresponding *Supporting information*.

## ADDITIONAL FILES

The following material is available online.

### Supplemental Material

**Supplemental material (Spectrum00954-23-s0001.docx).** Figures, tables, and spectra.
**Table S3 (Spectrum00954-23-s0002.xlsx).** Nature and resistance profiles of the three E. coli strains used in the study.

## Open Peer Review

**PEER REVIEW HISTORY (review-history.pdf).** An accounting of the reviewer comments and feedback.

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
