## [Reviewer comments · Microbiology Spectrum]

Microbiology Spectrum

In Vitro Antibacterial Activity of Dinuclear Thiolato-Bridged Ruthenium(II)-Arene Compounds

Quentin Bugnon, Camilo Melendez, Oksana Desiatkina, Louis Fayolles Chorus de Chaptès, Isabelle Holzer, Paunescu Emilia, Markus Hilty, and Julien Furrer

Corresponding Author(s): Julien Furrer, Universitat Bern

Review Timeline:

Submission Date:	May 19, 2023
Editorial Decision:	July 5, 2023
Revision Received:	August 2, 2023
Accepted:	August 28, 2023

Editor: Silvia Cardona

Reviewer(s): The reviewers have opted to remain anonymous.

Transaction Report:

DOI: <https://doi.org/10.1128/spectrum.00954-23>

July 5, 2023

Prof. Julien Furrer
Universitat Bern
Department of Chemistry, Biochemistry and Pharmaceutical Sciences
Freiestrasse 3
Berne 3012
Switzerland

Re: Spectrum00954-23 (In Vitro Antibacterial Activity of Dinuclear Thiolato-Bridged Ruthenium(II)-Arene Compounds)

Dear Prof. Julien Furrer:

Thank you for submitting your manuscript to Microbiology Spectrum. Your manuscript has been reviewed by two experts in the field. Both have major recommendations to be addressed before the manuscript can be considered for publication. Specifically, one reviewer stated that the conclusion related to the ICP-MS results is not supported by the data presented. Another reviewer was concerned that your work has not included any data on possible toxicity. Please, ensure you address these comments if you decide to submit a revision.

Link Not Available

Sincerely,

Silvia Cardona

Journals Department
Reviewer comments:

Reviewer #1 (Comments for the Author):

This manuscript evaluated the activity of 22 Ru complexes against *E. coli*, *S. pneumoniae* and *S. aureus*. The new compounds synthesized were thoroughly characterized by both HRMS and elemental analysis to ensure high purity. Several compounds showed antibacterial activity against *S. pneumoniae* and a few others were active against *S. aureus*, with MICs in low μM . The

authors observed an overall decrease of the bactericidal effect with the increase of molecular weight, and the complexes bearing larger benzolactam had low MICs.

1. Are the strains MDR? The nature and resistance profiles of the strains should be included, as the motivation of the work is to combat MDR.
2. Stability studies were done in DMSO rather than biologically relevant solvents like buffers or growth media. It is true that the compounds are insoluble in aqueous solutions, but one could perhaps test stability using a mixed solvent like DMSO-d₆/D₂O.
3. Vague descriptive words should be avoided and substituted with more precise descriptions. For example, in "...very limited solvolysis of the ester bonds...", what constitutes "limited solvolysis"? What percentage of the ester bonds were hydrolyzed?
4. The quality of Figure 6 is very poor. The fluorescence was not visible in any of the images, and no cells were seen at either the white or green arrows.
5. In the cellular uptake experiments, after incubating with bacteria, the Ru compounds were washed with aq. 0.85% NaCl twice. Since the Ru compounds are insoluble in aqueous solution, this procedure will not remove physisorbed compounds on the bacteria, thus leaving residual compounds on the bacteria. This may explain why there was no dependence of the uptake on the incubation time. Correct washing procedure should be used to ensure that the ICP-MS results are indeed from internalized compounds.
6. The purity of commercial antibiotics should be included.

Reviewer #2 (Comments for the Author):

review attached

Staff Comments:

Preparing Revision Guidelines

Please return the manuscript within 60 days; if you cannot complete the modification within this time period, please contact me. If you do not wish to modify the manuscript and prefer to submit it to another journal, please notify me of your decision immediately so that the manuscript may be formally withdrawn from consideration by Microbiology Spectrum.

Comments on Spectrum00954-23

Overall:

This manuscript reports on the preparation of a series of diruthenium organometallic complexes employing a variety of functionalized ligands including BODIPY fluorescent probes and some lactam containing rings.

Overall the synthesis of the organometallics is carried out according to synthetic procedures that are standard in this area and the characterization data appears to be consistent with the proposed structures.

The rationale for this entire work is not fully clear — other than *in vitro* testing of some organometallics. It is far from clear if these molecules would have any utility as a clinical therapeutic. Glaring in the absence is the inclusion of any discussion of the potential human toxicity of ruthenium complexes and the barrier that might generate to any further utility of these molecules.

A major scientific shortcoming of the work reported in the manuscript is any preliminary testing on mammalian cell lines. At a minimum a test in a cell line such as HEK293 would potentially provide some insight into the utility of these organometallic complexes. The time and expenses of mammalian cell line testing is acknowledged but in the case of these organometallics it would be necessary to ensure that this work is viewed with the appropriate impact. A conclusive demonstration of selectivity for bacterial cells over mammalian would certainly provide some helpful rationale for this project.

At present this work offers little of significant impact and would not generate much interest in the medicinal chemistry community.

Specific comments:

Complex stability: the authors carry out a stability study by leaving samples in DMSO- d_6 — however it is not clear if this was left in an NMR tube and how it was stored for the intervening 100 days. This is also not a very relevant stability test and perhaps a test with at least some concentration of water (*not* D_2O) present in the DMSO would be more biologically relevant

NMR Characterization: Data: In the supporting information the authors report the ^{13}C -NMR chemical shifts to 2 decimal places whereas one decimal place is more appropriate even at 100 MHz. Unless the authors have made changes to the default Bruker settings for collecting a broad-band decoupled spectrum then these should be indicated.

Prof. Dr. Julien Furrer

Freiestrasse 3
CH - 3012 Bern

Tel: +41 031 684 43 83

julien.furrer@unibe.ch
<http://furrer.dcb.unibe.ch/>

^b
UNIVERSITÄT
BERN

Prof. Silvia Cardona
Editor *Microbiology Spectrum*

Monday, August 28, 2023

Submission of a revised manuscript Spectrum00954-23 for *Microbiology Spectrum*

Dear Prof Cardona,

Thank you for considering our manuscript for *Microbiology Spectrum* and for giving us the opportunity to respond and modify our manuscript.

The authors wish to thank the reviewers for their constructive comments that help to improve the quality of the manuscript. We have revised our original manuscript according to their recommendations/comments as detailed below.

Reviewer #1:

This manuscript evaluated the activity of 22 Ru complexes against *E. coli*, *S. pneumoniae* and *S. aureus*. The new compounds synthesized were thoroughly characterized by both HRMS and elemental analysis to ensure high purity. Several compounds showed antibacterial activity against *S. pneumoniae* and a few others were active against *S. aureus*, with MICs in low μ M. The authors observed an overall decrease of the bactericidal effect with the increase of molecular weight, and the complexes bearing larger benzolactam had low MICs.

1. Are the strains MDR? The nature and resistance profiles of the strains should be included, as the motivation of the work is to combat MDR.

- *The p53.1R and P54.2R strains were esbl while the P54.1T was colistin resistant. We have included a table summarizing the nature and resistance profiles of the three strains in the Supplementary Information.*

2. Stability studies were done in DMSO rather than biologically relevant solvents like buffers or growth media. It is true that the compounds are insoluble in aqueous solutions, but one could perhaps test stability using a mixed solvent like DMSO-d₆/D₂O.

We know that those trithiolato-bridged ruthenium(II)-arene compounds are very stable in water / biological media (see our previous reports, J. Furrer Inorg. Chem., 50 (2011), p. 10552, J. Biol. Inorg. Chem. 2012, 951, New J. Chem. 2013, 37, 3503–3511, Coord. Chem. Rev. 2016, 309, 36) and in DMSO; – Compounds possessing potentially hydrolysable ester bonds are also very stable, as shown in ChemBioChem, 21 (2020) 2818-2835 and Pharmaceuticals, 13 (2020). The ¹H-NMR spectra of BODIPY compounds provided in ChemBioChem, 23, e202200536, 2022 (compounds 13-16) or in Journal of Organometallic Chemistry 2023,

986:122624 (compounds 9-10) show that there is already a significant amount of water (H₂O) in the NMR tube.

3. Vague descriptive words should be avoided and substituted with more precise descriptions. For example, in "...very limited solvolysis of the ester bonds...", what constitutes "limited solvolysis"? What percentage of the ester bonds were hydrolyzed?

- *We agree with this referee that the phrase wording is not optimal. However, it is very difficult to provide precise percentage values. Previous NMR & Fluorescence spectra on many of the reported compounds indeed show a very limited solvolysis (~2%). As reported in ChemBiochem, 21 (2020) 2818-2835 and in ChemBioChem, 23, e202200536, 2022, BODIPY and coumarin conjugates always exhibited a small bathochromic shift of the fluorescence maximum in EtOH compared to CHCl₃ ($\Delta\lambda$ ca. 10 nm), and the intense fluorescence intensity change depending on the presence, or not, of the trithiolato di-ruthenium unit could be used to monitor the stability of the conjugates towards hydrolysis and their behavior in vitro.*

We have therefore reworded this sentence: "... and for these compounds, fluorescence measurements proved that the ester bonds remained intact for 48 h, but that a very limited hydrolysis of the ester bonds with the release of coumarin dyes after 168 h could be observed".

4. The quality of Figure 6 is very poor. The fluorescence was not visible in any of the images, and no cells were seen at either the white or green arrows.

- *We regret that the quality of figure 6 is poor. However, this is not due to the figure itself, but rather to the eps <-> pdf conversion. In fact, the original figure is of good quality and all the elements required for its interpretation are clearly visible.*

5. In the cellular uptake experiments, after incubating with bacteria, the Ru compounds were washed with aq. 0.85% NaCl twice. Since the Ru compounds are insoluble in aqueous solution, this procedure will not remove physisorbed compounds on the bacteria, thus leaving residual compounds on the bacteria. This may explain why there was no dependence of the uptake on the incubation time. Correct washing procedure should be used to ensure that the ICP-MS results are indeed from internalized compounds.

- *The water solubility of the Ru-compounds investigated, while generally low, varies widely from one compound to another. Indeed, the solubility of the unsubstituted Ru-compounds 5 & 6 can reach 0.5 mM (as reported in one of our first studies, ref 38, Inorg. Chem, 2011, 50, 10552), while the solubility of compounds 9-10 bearing an alkyl chain or conjugates 13-16 having a BODIPY group is effectively almost zero. As a result, we do not believe that hypothetical physisorbed compounds may explain why we did not observe a dependence between the uptake and the incubation time. For instance, the ICPMS provided the same value for the moderately water-soluble compound 5 and the insoluble BODIPY compound 15. Even if we suppose that remains of Ru-compounds were physisorbed to the bacteria, the bacteria, when lysed, must have been broken into fragments that are small enough to pass through the filters we used (0.22 μ m Millipore syringe filters), and this should have been detected. As such, in our opinion, we don't think we should modify the washing procedure.*

6. The purity of commercial antibiotics should be included.

- *The purity of the antibiotics has been added.*

Reviewer #2:

This manuscript reports on the preparation of a series of diruthenium organometallic complexes employing a variety of functionalized ligands including BODIPY fluorescent probes and some lactam containing rings. Overall the synthesis of the organometallics is carried out according to synthetic procedures that are standard in this area and the characterization data appears to be consistent with the proposed structures. The rationale for this entire work is not fully clear – other than in vitro testing of some organometallics. It is far from clear if these molecules would have any utility as a clinical therapeutic. Glaring in the absence is the inclusion of any discussion of the potential human toxicity of ruthenium complexes and the barrier that might generate to any further utility of these molecules. A major scientific shortcoming of the work reported in the manuscript is any preliminary testing on mammalian cell lines. At a minimum a test in a cell line such as HEK293 would potentially provide some insight into the utility of these organometallic complexes. The time and expenses of mammalian cell line testing is acknowledged but in the case of these organometallics it would be necessary to ensure that this work is viewed with the appropriate impact. A conclusive demonstration of selectivity for bacterial cells over mammalian would certainly provide some helpful rationale for this project. At present this work offers little of significant impact and would not generate much interest in the medicinal chemistry community.

- *We thank this referee for his critical but fair assessment of the submitted version. We agree that providing existing data on the toxicity against Human Foreskin Fibroblasts (HFFs) of these compounds (published in Eur J Med Chem **2021**, 222:113610, ChemBioChem, 23, e202200536, **2022**, Molecules, 28, 902, **2023**, Journal of Organometallic Chemistry **2023**, 986:122624. In summary: Compounds 3, 5, 6, 12, 13, 15, 16 were ranked as very promising, with no toxic effect on HFF cells at a high concentration of 2.5 μ M, compounds 1, 2, 7, 8 were moderately toxic, and compounds 9 and 10 were very toxic on HFF. We have compiled the data in a table included in the supplementary material (table S1) and added a paragraph in the main text mentioning toxicity and selectivity.*
- *Evaluation of the toxicity of compounds 17-22 against HEK293 cells has been performed (manuscripts still under preparation). The IC₅₀ values range between 0.18 and 7.2 μ M, and the selectivity Index (IC₅₀ HEK293 / IC₅₀ A24) between 1.2 and 3.*

Specific comments:

Complex stability: the authors carry out a stability study by leaving samples in DMSO-d₆ – however it is not clear if this was left in an NMR tube and how it was stored for the intervening 100 days. This is also not a very relevant stability test and perhaps a test with at least some concentration of water (not D₂O) present in the DMSO would be more biologically relevant.

- *The samples dissolved in DMSO were left in the NMR tube and stored in the laboratory at room temperature, or at 4°C in the dark (BODIPY-compounds). Concerning the presence of a small amount of H₂O, the ¹H-NMR spectra of BODIPY compounds provided in ChemBioChem, 23, e202200536, **2022** (compounds 13-16) or in Journal of Organometallic Chemistry **2023**, 986:122624 (compounds 9-10) show that there is already a non negligible amount of water (H₂O) in the NMR tube.*

NMR Characterization: Data: In the supporting information the authors report the ¹³C- NMR chemical shifts to 2 decimal places whereas one decimal place it more appropriate even at 100 MHz. Unless the authors have made changes to the default Bruker settings for collecting a broad-band decoupled spectrum then these should be indicated.

- *We thank this referee for pointing out this point. Our ^{13}C -NMR spectra are recorded with 64k points (standard acquisition parameters, acquisition time of 1.2s), leading to a resolution (Hz per point) of 0.5 Hz, thus 0.005 ppm. It is therefore possible to give an accuracy of 0.01 ppm. For example, for compound 17, in the ^{13}C spectrum, the two carbons at 143.97 and 143.92 ppm are perfectly discernible. As many of the reported carbon provide NMR resonances very close to each other, we prefer to keep the chemical shifts with a precision of 2 decimals.*

We thank you for your attention to this manuscript.

Sincerely,

Julien Furrer, Prof. Dr.

August 28, 2023

Prof. Julien Furrer
Universitat Bern
Department of Chemistry, Biochemistry and Pharmaceutical Sciences
Freiestrasse 3
Berne 3012
Switzerland

Re: Spectrum00954-23R1 (In Vitro Antibacterial Activity of Dinuclear Thiolato-Bridged Ruthenium(II)-Arene Compounds)

Dear Prof. Julien Furrer:

Your manuscript has been accepted, and I am forwarding it to the ASM Journals Department for publication. You will be notified when your proofs are ready to be viewed.

Sincerely,

Silvia Cardona
Editor, Microbiology Spectrum
